# Contextual Pricing for Lipschitz Buyers

**Jieming Mao**
University of Pennsylvania
jiemingm@seas.upenn.edu

**Renato Paes Leme**
Google Research
renatoppl@google.com

**Jon Schneider**
Google Research
jschnei@google.com

## Abstract

We investigate the problem of learning a Lipschitz function from binary feedback. In this problem, a learner is trying to learn a Lipschitz function $f : [0, 1]^d \to [0, 1]$ over the course of $T$ rounds. On round $t$, an adversary provides the learner with an input $x_t$, the learner submits a guess $y_t$ for $f(x_t)$, and learns whether $y_t > f(x_t)$ or $y_t \le f(x_t)$. The learner's goal is to minimize their total loss $\sum_t \ell(f(x_t), y_t)$ (for some loss function $\ell$). The problem is motivated by *contextual dynamic pricing*, where a firm must sell a stream of differentiated products to a collection of buyers with non-linear valuations for the items and observes only whether the item was sold or not at the posted price.

For the symmetric loss $\ell(f(x_t), y_t) = |f(x_t) - y_t|$, we provide an algorithm for this problem achieving total loss $O(\log T)$ when $d = 1$ and $O(T^{(d-1)/d})$ when $d > 1$, and show that both bounds are tight (up to a factor of $\sqrt{\log T}$). For the pricing loss function $\ell(f(x_t), y_t) = f(x_t) - y_t \mathbf{1}\{y_t \le f(x_t)\}$ we show a regret bound of $O(T^{d/(d+1)})$ and show that this bound is tight. We present improved bounds in the special case of a population of linear buyers.

## 1 Introduction

A major problem in revenue management is designing pricing strategies for highly differentiated products. Besides the usual tension between exploration and exploitation (often call learning and earning in revenue management) the problem poses the following additional challenges: (i) the feedback in pricing problems is very limited: for each item the seller only learns whether the item was sold or not; (ii) the loss function is discontinuous and asymmetric: pricing slightly under the buyer's valuation causes a small loss while pricing slightly above causes the item not to be sold and therefore a large loss.

The study of learning in pricing settings was pioneered by Kleinberg and Leighton [15] who designed optimal pricing policies in a variety of settings when the products are undifferentiated. Motivated by applications to online commerce and internet advertisement, there has been a lot of interest in extending such results to contextual settings, where the seller is able to observe characteristics of each product, typically encoded by a high-dimensional feature vector $x_t \in \mathbb{R}^d$. The typical approach in those problems has been to assume that the valuation of the buyer is linear (Amin et al [2], Cohen et al [10], Lobel et al [20], Nazerzadeh and Javanmard [14], Javanmard [13] and Paes Leme and Schneider [19]) or that the demand function of a population of buyers is linear (Qiang and Bayati [21]).

Here we focus on the cases where the buyer's valuation is non-linear in the feature vectors, or where there are multiple buyers all with linear valuation functions. These cases can be cast as special cases of the semi-Lipschitz bandits model of Cesa-Bianchi et al [8]. Our goal is to exploit the special structure of the pricing problem and obtain improved bounds compared to those achieved for semi-Lipschitz bandits.

The model is as follows: our seller receives a new item for each of $T$ rounds. The item at time $t$ is described by a feature vector $x_t \in \mathbb{R}^d$. The seller is selling these items to a population of $b$ buyers, where buyer $i$ is willing to pay up to $V_i(x_t)$ for item $x_t$ (for some valuation function $V_i$ unknown to the seller). Every round the seller gets to choose a price $p_t$ for the current item. If $p_t \leq V_i(x_t)$ for some $i$, then some buyer purchases the item and the seller receives revenue $p_t$. Otherwise, no buyer purchases the item and the seller receives revenue $0$. The goal of the seller is to maximize their revenue, and in particular minimize the difference between their revenue and the revenue of a seller who knows the $V_i$'s ahead of time (their *regret*).

For the special case where there is a single buyer ($b = 1$) and his valuation is linear in $x_t$, the tight bound of $O(\mathrm{poly}(d) \log \log T)$ was recently given in [19]. In this paper, we consider the setting where the number of buyers $b$ is very large (potentially infinite), and we want regret bounds independent of $b$. We show:

- If all the $V_i$ are $L$-Lipschitz, then there is an algorithm for this contextual pricing problem that achieves regret $\Theta((LT)^{d/(d+1)})$, which is tight (Theorems 4, 7). This improves over the $O(T^{(d+1)/(d+2)})$ bound that we obtain by applying the algorithms for semi-Lipschitz bandits [8].

- If all the $V_i$ are linear (i.e. of the form $V_i(x) = \langle v_i, x \rangle$ for some $v_i \in [0,1]^d$), then there is an algorithm for this contextual pricing problem that achieves regret $O_d(T^{(d-1)/d})$ (Corollary 11). We exploit the special structure by casting this pricing problem as learning the extreme points of a convex set from binary feedback. We also show that any algorithm for this problem must incur regret at least $\Omega_d(T^{(d-1)/(d+1)})$ (Theorem 12). The lower bound is obtained through a connection to spherical codes.

To prove these results, we investigate a more general problem, which we term *learning a Lipschitz function with binary feedback*, and which may be of independent interest. In this problem, a learner is trying to learn an $L$-Lipschitz function $f : [0,1]^d \to [0,1]$ over the course of $T$ rounds. On round $t$, the learner is (adversarially) provided with a *context* $x_t$; the learner must then submit a guess $y_t$ for $f(x_t)$, upon which they learn whether $y_t > f(x_t)$ or $y_t \leq f(x_t)$ (and notably, not the value of $f(x_t)$). The learner's goal is to minimize their total loss $\sum_t \ell(f(x_t), y_t)$, for some loss function $\ell(\cdot, \cdot)$.

For the symmetric loss function $\ell(\theta, y) = |\theta - y|$ we provide the following regret bounds:

- when $d = 1$, there is an algorithm which achieves regret $O(L \log T)$ (Theorem 2). Any algorithm for this problem must incur regret $\tilde{\Omega}(L\sqrt{\log T})$ (Theorem 8).

- for $d > 1$, there is an algorithm which achieves regret $\Theta(LT^{(d-1)/d})$, which is tight (Theorems 3, 6).

We note that our problem for the symmetric loss function is no longer an instance of Lipschitz or semi-Lipschitz bandits, since the feedback is very restricted: the algorithm doesn't learn the actual loss – it only receives binary feedback as to whether its guess was above or below the true value.

We present two types of algorithms for this problem. The first set of algorithms are based around the divide-and-conquer strategy of *iterative partition refinement* which is the main workhorse for dealing with Lipschitz assumptions in learning [18, 17, 23, 12]. Here the algorithm starts with a partition of the domain of $f$ (perhaps just the domain itself), and tries to approximate $f$ on each element of this partition. When the algorithm approximates $f$ on a given element of the partition accurately enough, it further divides that element.

The second set of algorithms does not keep track of a partition of the domain but instead maintains lower and upper estimates of the function we are trying to learn. For example, we show that the natural algorithm which simply chooses the point halfway between the smallest possible value of $f(x_t)$ and the largest possible value of $f(x_t)$ consistent with the information known so far (the "midpoint algorithm") also achieves our optimal regret bounds. Such algorithms have the advantage that information learned about $f(x_t)$ is not necessarily confined to points in the vicinity of $x_t$, and thus may perform better in practice. See Section 2.3 for details.

Our lower bounds largely follow directly from the analysis of our algorithms, with the notable exception of the $\Omega(\sqrt{\log T})$ lower bound for the symmetric loss when $d = 1$. To prove this lower

bound, we demonstrate how to construct a family of Lipschitz functions which encode random walks of length $\approx \log T$ in the slopes between queried points. Understanding how to close the gap between $\Omega(\sqrt{\log T})$ and $O(\log T)$ for this case is an interesting open question.

The remainder of this paper is organized as follows. In the rest of this section, we discuss related work and formally define the problem of learning a Lipschitz function with binary feedback. In Section 2, we present our algorithms for learning a Lipschitz function with binary feedback, and in Section 3, we provide corresponding lowerbounds. Finally, in Section 4, we discuss how to apply these results to the contextual pricing problem (with emphasis on the setting with multiple linear buyers). For conciseness, the majority of proofs are omitted from the main body and appear in the appendix of the Supplementary Material.

## 1.1 Related Work

Our work belongs to the intersection of two major streams of literature: (i) learning for revenue optimization and (ii) continuum-armed and Lipschitz bandits. For revenue optimization, besides the work on contextual learning cited earlier, there are interesting other interesting directions such a learning with limited inventory. See for example Besbes and Zeevi [5], Babaioff et al [3], Badanidiyuru et al [4], Wang et al [24] and den Boer and Zwart [11]. Also relevant is the work on learning parametric models: Broder and Rusmevichientong [7], Chen and Farias [9] and Besbes and Zeevi [6].

Another relevant line of work is research on continuum-armed and Lipschitz bandits. The problem was introduced by Agrawal [1] and nearly tight bounds were obtained by Kleinberg [18]. Later the model was been extended to general metric spaces by Slivkins [22], Kleinberg and Slivkins [16] and Kleinberg, Slivkins and Upfal [17]. The problem with similarity information on contexts is studied by Hazan and Megiddo [12]. Slivkins [23] extends the Lipschitz bandits to contextual settings, i.e., when there is similarity information on both contexts and arms. Cesa-Bianchi et al [8] study the problem under partial feedback and weaken the Lipschitz assumption in previous work to semi-Lipschitz.

## 1.2 Learning a Lipschitz function from binary feedback

**Definition 1.** *A function $f : \mathbb{R} \to \mathbb{R}$ is L-Lipschitz if, for all $x, y \in \mathbb{R}$, $|f(x) - f(y)| \leq L|x - y|$.*

In this paper we study the problem of *learning a Lipschitz function from binary feedback*. This problem can be thought of as the following game between an adversary and a learner. At the beginning, the adversary chooses an $L$-Lipschitz function $f : [0, 1] \to [0, 1]$. Then, on round $t$ (for $T$ rounds), the adversary begins by providing the learner with a point $x_t \in [0, 1]$. The learner must then submit a guess $y_t$ for $f(x_t)$. The learner then learns whether $y_t > f(x_t)$ or not. The goal of the learner is to minimize their *total loss* (alternatively, *regret*) over $T$ rounds, $\mathsf{Reg} = \sum_{t=1}^{T} \ell(f(x_t), y_t)$, where $\ell(\cdot, \cdot)$ is some loss function.

In this paper, we consider the following two loss functions:

**Symmetric loss.** The symmetric loss is given by the function $\ell(f(x_t), y_t) = |f(x_t) - y_t|$. This is simply the distance between the learner's guess and the true value.

**Pricing loss.** The pricing loss is given by the function $\ell(f(x_t), y_t) = f(x_t) - y_t \mathbf{1}\{y_t \leq f(x_t)\}$. In other words, the pricing loss equals the symmetric loss when the guess $y_t$ is less than $f(x_t)$ (and goes to 0 as $y_t \to f(x_t)^-$), but equals $f(x_t)$ when the guess $y_t$ is larger than $f(x_t)$. This loss often arises in pricing applications (where setting a price slightly larger than optimal leads to no sale and much higher regret than a price slightly lower than optimal).

We also consider a variant of this problem for higher-dimensional Lipschitz functions. For functions $f : \mathbb{R}^d \to \mathbb{R}$, we define $L$-Lipschitz with respect to the $L_\infty$-norm on $\mathbb{R}^d$: $|f(x) - f(y)| \leq L\|x - y\|_\infty$ for all $x, y \in \mathbb{R}^d$. Our results hold for other $L_p$ norms on $\mathbb{R}^d$, up to polynomial factors in $d$. We can then define the problem of learning a (higher-dimensional) Lipschitz function $f : [0, 1]^d \to [0, 1]$ analogously as to above.

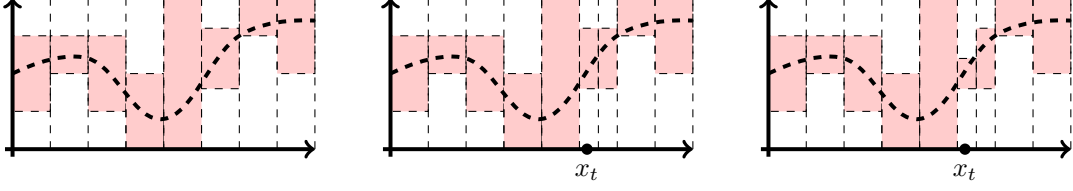

Figure 1: Illustration of Algorithm 1: the dashed curve corresponds to the (unknown) Lipschitz function, the rectangles correspond to feasible regions for the function. When an update results in a part of the partition with small relative height, we bisect this part of the partition.

Oftentimes, we will want to think of $d$ as fixed, and consider only the asymptotic dependence on $T$ of some quantity (e.g. the regret of some algorithm). We will use the notation $O_d(\cdot)$ and $\Omega_d(\cdot)$ to hide the dependency on $d$.

## 2 Algorithms for learning a Lipschitz function

### 2.1 Symmetric Loss

In this subsection we present algorithms for learning Lipschitz functions under the symmetric loss that incur sublinear total regret. Without loss of generality, we will assume in this section that $L \geq 1$ (the results in the appendix of the supplementary material allow us to extend these algorithms to $L \leq 1$ with slight modifications to the regret bounds).

We begin by examining the case where $d = 1$ (the functions are from $\mathbb{R} \to \mathbb{R}$). The following algorithm (Algorithm 1) achieves total loss $O(L \log T)$. Algorithm 1 maintains a partition of the domain of $f$ ($[0, 1]$) into a collection of intervals $X_j$. For each interval $X_j$, the algorithm maintains an associated interval $Y_j$ that satisfies $f(X_j) \subseteq Y_j$.

When a point $x$ in $X_j$ is queried, the learner submits as their guess the midpoint $y$ of the interval $Y_j$. The binary feedback of whether $y > f(x)$ or not allows the learner to update the interval $Y_j$, shrinking it. Once $Y_j$ grows small enough with respect to $X_j$, we bisect $X_j$ into two smaller intervals. This procedure is illustrated in Figure 1.

---

**Algorithm 1** Algorithm for learning a $L$-Lipschitz function from $\mathbb{R}$ to $\mathbb{R}$ under symmetric loss with regret $O(L \log T)$.

---

1: Learner maintains a partition of $[0, 1]$ into intervals $X_j$.
2: Along with each interval $X_j$, learner maintains an associated range $Y_j \subseteq [0, 1]$ such that if $x \in X_j$, $f(x) \in Y_j$.
3: Initially, learner partitions $[0, 1]$ into $\lceil 8L \rceil$ intervals $X_j$ of equal length $\leq 1/8L$ and sets all $Y_j = [0, 1]$.
4: **for** $t = 1$ to $T$ **do**
5:      Learner receives an $x_t \in [0, 1]$ from the adversary.
6:      Learner finds $j$ s.t. $x_t \in X_j$. Let $\ell_j = \text{length}(X_j)$.
7:      Learner guesses $y_t = (\max(Y_j) + \min(Y_j))/2$.
8:      **if** $y_t > f(x_t)$ **then**
9:          $Y_j \leftarrow Y_j \cap [0, y_t + L\ell_j]$
10:     **else**
11:         $Y_j \leftarrow Y_j \cap [y_t - L\ell_j, 1]$.
12:     **end if**
13:     Let $h_j = \text{length}(Y_j)$.
14:     **if** $h_j < 4L\ell_j$ **then**
15:         Bisect $X_j$ to form two new intervals $X_{j_1}$ and $X_{j_2}$. Set $Y_{j_1} = Y_{j_2} = Y_j$.
16:     **end if**
17: **end for**

---

**Theorem 2.** *Algorithm 1 achieves regret $O(L \log T)$ for learning a $L$-Lipschitz function with symmetric loss.*

Roughly, the proof of Theorem 2 follows from the following two properties: i) after a constant number of queries belonging to any interval $X_j$, the interval $Y_j$ will shrink enough to trigger a bisection, and ii) the regret of a query in an interval $X_j$ is at most $\mathsf{length}(Y_j)$ which itself is $O(\mathsf{length}(X_j))$.

Now, if we start with $\Theta(1)$ intervals of length $\Theta(1)$, throughout the process there will be at most $O(2^r)$ intervals of length $\Theta(2^{-r})$ (those intervals bisected $r$ times). Since each query in an interval of length $\ell$ contributes $O(\ell)$ to the overall regret, this means that the total regret from $T$ queries is at most $O(1 + 2 \cdot 2^{-1} + 2^2 \cdot 2^{-2} + \cdots + 2^{\log T} \cdot 2^{-\log T}) = O(\log T)$.

It is possible to extend Algorithm 1 (in a straightforward way) to Lipschitz functions from $\mathbb{R}^d$ to $\mathbb{R}$. Pseudocode for this algorithm is provided in the appendix of the supplementary material. Here, for $d > 1$, we no longer get logarithmic regret; instead, our algorithm achieves regret $O(LT^{(d-1)/d})$.

**Theorem 3.** *There exists an algorithm that achieves regret $O(LT^{(d-1)/d})$ for learning a L-Lipschitz function from $\mathbb{R}^d$ to $\mathbb{R}$ with symmetric loss.*

The main difference between Theorem 3 and Theorem 2 is that there are now $O(2^{dr})$ "intervals" ($d$-dimensional boxes) of diameter $\Theta(2^{-r})$, so the total regret from $T$ queries is now $O(1 + 2^d \cdot 2^{-1} + 2^{2d} \cdot 2^{-2} + \cdots + 2^{\log T} \cdot 2^{-(\log T)/d}) = O(T^{(d-1)/d})$.

## 2.2  Pricing Loss

We now explore algorithms that achieve low regret with respect to the pricing loss function. Our main approach will be to adapt our algorithm from Theorem 3 (which achieves low regret with respect to the symmetric loss function for Lipschitz functions from $\mathbb{R}^d$ to $\mathbb{R}$) but stop subdividing once the length of a range $Y_j$ drops below some threshold. The details are summarized in the appendix of the supplementary material.

We show that our algorithm achieves regret $O((LT)^{d/(d+1)})$. Note that for $d = 1$, this is $O(L\sqrt{T})$; unlike in the symmetric loss case, it is impossible to achieve logarithmic regret for the pricing loss (see Theorem 7).

**Theorem 4.** *There exists an algorithm that achieves regret $O((LT)^{d/(d+1)})$ for learning a L-Lipschitz function from $\mathbb{R}^d$ to $\mathbb{R}$ with pricing loss.*

As with Theorem 3, a similar analysis to that of Theorem 2 holds, with the exception that the regret of a query in an interval is $O(1)$ (until the length of the interval shrinks below some threshold, in which case we play $\min Y_j$ and are guaranteed regret at most $\mathsf{length}(Y_j)$). Choosing this threshold optimally results in the above regret bound.

## 2.3  Midpoint algorithms

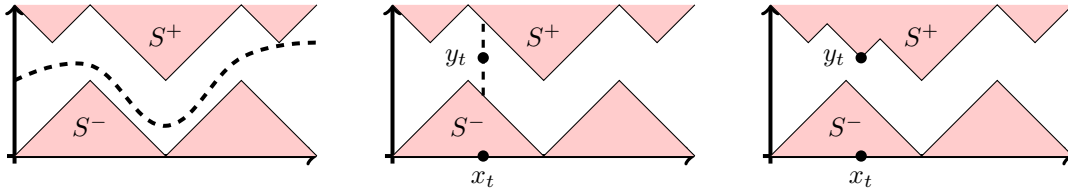

Figure 2: Illustration of the Midpoint Algorithm (Algorithm 2).

Let us return to considering the one-dimensional instance of learning an $L$-Lipschitz function under the symmetric loss. One very natural algorithm for this problem is the following. Throughout the algorithm, maintain two subregions of $[0, 1]^2$; $S^+$, a set of points $\{(x, y)\}$ that we know are guaranteed to satisfy $y \geq f(x)$, and $S^-$, a set of points $\{(x, y)\}$ that we know are guaranteed to satisfy $y \leq f(x)$.

Initially, $S^+$ and $S^-$ start empty (or more accurately, containing the two lines $[0, 1] \times \{1\}$ and $[0, 1] \times \{0\}$, respectively). Each time we receive feedback of the form $y_t > f(x_t)$, we can add all points $(x, y)$ which satisfy $y \geq y_t + L|x_t - x|$ to $S^+$; by the $L$-Lipschitz condition, all such points

satisfy $y \geq f(x)$. Similarly, each time we receive feedback of the form $y_t < f(x_t)$, we can add all points $(x, y)$ which satisfy $y \leq y_t - L|x_t - x|$ to $S^-$.

Finally, to choose $y_t$ given $x_t$, we should choose some $y_t$ between $y^- = \max\{y|(x_t, y) \in S^-\}$ and $y^+ = \min\{y|(x_t, y) \in S^+\}$. A natural choice is their midpoint $(y^- + y^+)/2$. We call this algorithm the *midpoint algorithm*; its details are summarized in Algorithm 2. This process is depicted in Figure 2.

---

**Algorithm 2** Midpoint algorithm for learning a $L$-Lipschitz function from $\mathbb{R}$ to $\mathbb{R}$ under symmetric loss with regret $O(L \log T)$.

---

1: Learner maintains two polygonal subsets $S^+$ and $S^-$ of $[0,1]^2$.
2: Initially, $S^+ = \{(x, 1)|x \in [0, 1]\}$ and $S^- = \{(x, 0)|x \in [0, 1]\}$.
3: **for** $t = 1$ to $T$ **do**
4:     Learner receives an $x_t \in [0, 1]$ from the adversary.
5:     Learner computes $y^- = \max\{y|(x_t, y) \in S^-\}$ and $y^+ = \min\{y|(x_t, y) \in S^+\}$.
6:     Learner guesses $y_t = (y^- + y^+)/2$.
7:     **if** $y_t > f(x_t)$ **then**
8:         $S^+ \leftarrow S^+ \cup \{(x, y)|y \geq y_t + L|x_t - x|\}$.
9:     **else**
10:        $S^- \leftarrow S^- \cup \{(x, y)|y \leq y_t - L|x_t - x|\}$.
11:     **end if**
12: **end for**

---

Note that while Algorithm 1 and its variants are low-regret (with essentially tight matching lower-bounds) and efficiently implementable, they don't share information between different intervals $X_i$. One advantage of the midpoint algorithm over these algorithms is that information provided from a query at a point $x$ is not necessarily localized to the immediate neighborhood around $x$.

We show that, like Algorithm 1, the midpoint algorithm is also low regret.

**Theorem 5.** *Algorithm 2 achieves regret $O(\log T)$ for learning a L-Lipschitz function from $\mathbb{R}$ to $\mathbb{R}$ with symmetric loss.*

It is likewise possible to adapt the midpoint algorithm to multiple dimensions and to the pricing loss function (by choosing $y^-$ whenever $y^+ - y^-$ is below some threshold) and prove analogues of Theorems 3 and 4. We omit the details for conciseness.

## 3 Lower bounds for learning a Lipschitz function

In this section, we state lower bounds for our results in Section 2. Interestingly all our lower bounds also hold for a slightly easier problem in which the algorithm learns the value of $f(x_t)$ after round $t$ (instead of just whether $y < f(x_t)$).

Generally, all of our lower bounds work in the following way. We construct a collection $\mathcal{C}$ of $L$-Lipschitz functions and a sequence of queries $x_1, x_2, \ldots, x_T$ for the adversary such that for a random function $f$ in $\mathcal{C}$, $f(x_t)$ is equally likely to be $\frac{1}{2} + \delta_t$ or $\frac{1}{2} - \delta_t$ for some $\delta_t$, even conditioned on the values of $f(x_1)$ through $f(x_{t-1})$.

For both the symmetric loss when $d > 1$, and the pricing loss (for all $d$), constructing such a collection is easy; we simply divide the domain into small cubes, let $x_1$ through $x_T$ run over the centers of such cubes, and let $f(x_t)$ be either $\frac{1}{2} + \delta$ or $\frac{1}{2} - \delta$ with equal probability. Optimizing $\delta$ leads to the following tight lower bounds.

**Theorem 6.** *For $d > 1$ and $L \leq T^{1/d}$, any algorithm for learning an $L$-Lipschitz function with symmetric loss must incur $\Omega\left(LT^{\frac{d-1}{d}}\right)$ regret for the d-dimensional case.*

**Theorem 7.** *For $L \leq T^{1/d}$, any algorithm for learning an $L$-Lipschitz function with the pricing loss must incur $\Omega\left((LT)^{\frac{d}{d+1}}\right)$ regret for the d-dimensional case.*

More interesting is the case of the symmetric loss when $d = 1$. Here we obtain an $\tilde{\Omega}(\sqrt{\log T})$ lower bound.

**Theorem 8.** *Any algorithm for learning an L-Lipschitz function with symmetric loss must incur* $\Omega\left(L\sqrt{\frac{\log T}{\log\log T}}\right)$ *regret.*

The proof of Theorem 8 proceeds roughly as follows. Our queries $x_t$ will range over all the dyadic rationals, in order of increasing denominator (e.g. in the order $1/2, 1/4, 3/4, 1/8, 3/8, 5/8, 7/8$). We now use this sequence of $x_t$'s to adaptively construct a Lipschitz function $f(x)$ in the following way. We start by setting $f(0) = f(1) = 1/2$. To set the value of $f(x_t)$ for some $x_t = \frac{2i+1}{2^r}$, let $L = i/2^{r-1}$ and $R = (i+1)/2^{r-1}$ (note that $x_t$ is the midpoint of $[L, R]$, and $f(L)$ and $f(R)$ have already been chosen inductively). Let $m$ be the slope between $(L, f(L))$ and $(R, f(R))$. Now, we choose $f(x_t)$ so that the slope between $(L, f(L))$ and $(x_t, f(x_t))$ is $m + \delta$ with probability $1/2$, and $m - \delta$ with probability $1/2$. If this causes the Lipschitz condition to be violated (because $m + \delta > L$ or $m - \delta < -L$), we instead just set $f(x_t) = (f(L) + f(R))/2$.

This process has the interesting property that the slope of a segment of length $2^{-r}$ of this function $f$ is $\delta$ times a random walk of length $r$. If we choose $\delta = \Theta(1/\sqrt{\log T})$, then we can run this random walk for $\approx L \log T$ steps without running into this Lipschitz constraint (since the expected maximum value of a random walk of length $n$ is $\widetilde{\Theta}(\sqrt{n})$). Analyzing the regret for this choice of $\delta$ leads to the regret bound in Theorem 8. For more details, see the full paper.

## 4  Contextual Pricing for Linear Buyers

We now show how to apply our solutions to the problem of learning a Lipschitz function (in particular, with respect to the pricing loss function) to the problem of contextual dynamic pricing (with a particular emphasis on when all the buyers have linear valuation functions).

We begin by examining the case where each buyer $i$ (for $1 \leq i \leq b$) has an $L$-Lipschitz valuation function $V_i : [0,1]^d \to [0,1]$, with $V_i(x)$ representing how much they would be willing to pay for an item with features $x \in \mathbb{R}^d$. Let $f(x) = \max_i V_i(x)$. Note that the seller successfully makes a sale at round $t$ if $p_t \leq f(x_t)$, in which case the seller receives revenue $p_t$; otherwise, the seller receives revenue $0$. But now, note that since $f$ is the maximum of several $L$-Lipschitz functions, $f$ is also $L$-Lipschitz. This problem is therefore exactly the problem of learning a Lipschitz function with respect to the pricing loss function. Since $f$ can be any $L$-Lipschitz function from $[0,1]^d \to [0,1]$, lower bounds for learning such functions carry over to this dynamic pricing problem. Theorems 4 and 7 thus imply the following corollary.

**Corollary 9.** *There exists an algorithm for solving the contextual dynamic pricing problem for L-Lipschitz buyers in d dimensions with total regret $O((LT)^{d/(d+1)})$. Any algorithm for solving the contextual dynamic pricing problem for L-Lipschitz buyers in d dimensions must incur total regret at least $\Omega((LT)^{d/(d+1)})$.*

An interesting special case is the one where all buyers have linear valuations, i.e., $V_i(x) = \langle v_i, u \rangle$ for some vector $v_i \in [0,1]^d$. The case with $b = 1$ buyer is very well-studied and a regret bound of $O(\mathsf{poly}(d)\log\log T)$ is possible [19]. For $b > 1$, we exploit the special structure of the problem to improve over the $O(T^{d/(d+1)})$ guarantee of Corollary 9.

We begin by reinterpreting this problem geometrically as follows. Define $S$ to be the convex hull $\mathsf{conv}(0, v_1, v_2, \ldots, v_b)$. Note that there exists a buyer willing to buy an item $x_t \in [0,1]^d$ at price $p_t$ iff the hyperplane $\{u \in \mathbb{R}^d; \langle x_t, u \rangle = p_t\}$ intersects the set $S$. For this reason, we will abuse notation and refer to this convex hull $S$ as the "set of buyers" (indeed, adding a buyer with a $v$ corresponding to any point within $S$ does not change the outcome any sale). One can then alternatively view the dynamic pricing problem for linear buyers as the problem of learning the extreme points of a convex set $S \subseteq [0,1]^d$ from binary feedback.

In this problem, the context provided by the adversary is the feature vector $x_t$ of the item at time $t$. Since without loss of generality, this context $x_t$ is a unit vector in $\mathbb{R}^d$ (if it is not one, it can be scaled to become one along with the price, at the cost of at most a $\sqrt{d}$ factor in regret), and is therefore a $(d-1)$-dimensional object. We will parametrize these unit vectors via *generalized spherical coordinates*; that is, the $(d-1)$-tuple $(\theta_1, \theta_2, \ldots, \theta_{d-1}) \in [0, \pi/2]^{d-1}$ corresponds to the unit vector defined by

$$\left(\cos\theta_1, \sin\theta_1\cos\theta_2, \sin\theta_1\sin\theta_2\cos\theta_3, \ldots, \sin\theta_1\sin\theta_2\cdots\sin\theta_{d-2}\cos\theta_{d-1}\right).$$

Let $x(\theta)$ (for $\theta \in [0,\pi/2]^{d-1}$) be the above unit vector in $\mathbb{R}^d$. We make the following observation.

**Lemma 10.** *Let $F(\theta) = \max_{v \in S}\langle x(\theta), v\rangle$. Then $F$ is $L$-Lipschitz for $L = O(d^2)$.*

Now, note that the dynamic pricing problem for linear buyers is exactly the problem of learning the function $F$ with respect to the pricing loss; every round, the adversary supplies a context $\theta$, the seller submits a price $p$, and the seller receives revenue $p$ if $F(\theta) \geq p$ and revenue 0 otherwise. Theorem 4 immediately implies the following corollary.

**Corollary 11.** *There exists an algorithm for solving the contextual dynamic pricing problem in $d > 1$ dimensions with total regret $O(d^{2(d-1)/d}T^{(d-1)/d}) = O_d(T^{(d-1)/d})$.*

Unfortunately, not every Lipschitz function can occur as a valid $F(\theta)$, so the lower bounds from Section 3 do not immediately hold. Nonetheless, we can adapt the ideas from Theorem 7 to prove that any algorithm for solving the contextual dynamic pricing problem in $d$ dimensions must incur regret $\Omega_d(T^{(d-1)/(d+1)})$.

**Theorem 12.** *Any algorithm for solving the contextual dynamic pricing problem in $d > 1$ dimensions must incur total regret at least $\Omega_d(T^{(d-1)/(d+1)})$.*

To prove Theorem 12, we will need the following lemma regarding the maximum size of spherical codes.

**Lemma 13.** *Let $\alpha > 0$. Then there exists a set $U_\alpha$ of $\Theta_d(\alpha^{-(d-1)})$ unit vectors in $(\mathbb{R}^+)^d$ such that for any two distinct elements $u, u'$ of $U_\alpha$, $\langle u, u'\rangle \leq \cos\alpha$ (i.e. any two distinct unit vectors are separated by angle at least $\alpha$).*

We now proceed to prove Theorem 12.

*Proof of Theorem 12.* Choose $\alpha = \Theta_d(T^{-1/(d+1)})$. The adversary will choose the set $B$ of buyers as follows. For every element $v$ of the set $U_\alpha$ (defined in Lemma 13), the adversary with probability half adds $v$ to $B$, and otherwise adds $(\cos\alpha)v$ to $B$. The adversary will then choose the contexts as follows: for each element $u$ in $U_t$, the adversary will set $u_t = u$ for $T/|U_\alpha|$ rounds.

We claim no learning algorithm achieves $O_d(T^{(d-1)/(d+1)})$ regret against this adversary. Consider each element $u$ of $U_\alpha$, and consider the rounds where $x_t = u$. Either one of two cases must occur:

- **Case 1**: the algorithm never sets a price larger than $\cos\alpha$. Then, with probability $1/2$ (if $u \in B$), the maximal price the algorithm could have set was 1, so the algorithm incurs expected regret at least $\frac{1}{2}(1 - \cos\alpha)(T/|U_\alpha|) = \Omega_d\left(\alpha^2 \frac{T}{\alpha^{-(d-1)}}\right) = \Omega_d(T\alpha^{(d+1)}) = \Omega_d(1)$.

- **Case 2**: the algorithm at some point sets a price larger than $\cos\alpha$. Then, with probability $1/2$ (if $u \notin B$) the largest price the algorithm could have set was $\cos\alpha$ (since $\langle u', u\rangle \leq \cos\alpha$ for all other $u' \in u_t$, and we know $(\cos\alpha)u \in B$), so the algorithm overprices and incurs expected regret at least $\frac{1}{2}\cos\alpha = \Omega(1)$.

In either case, the algorithm incurs at least $\Omega_d(1)$ regret. Over all $|U_t|$ different contexts, this is at least $\Omega_d(T^{(d-1)/(d+1)})$ regret. $\square$

Closing the gap between the upper bound of $O_d(T^{(d-1)/d})$ and the lower bound of $\Omega_d(T^{(d-1)/(d+1)})$ is an interesting open problem.

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
