[Supplementary Material]

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

# A  Reductions Between Different Lipschitz Constants for Symmetric Loss

In this section, we provide black-box reductions that allow us to convert low-regret algorithms for learning $L$-Lipschitz functions into $L'$-Lipschitz functions. These reductions prove important for obtaining the correct regret bounds in $L$ in Section 2, along with allowing us to argue only about $L \geq 1$ (for the symmetric loss).

**Lemma 14.** *Suppose for some specific L, an algorithm A achieves regret $r(T)$ for learning an L-Lipschitz function with symmetric loss in the d-dimensional case, where $r(T)$ is concave in T. For any $L' > L$, let $\alpha = \lceil \frac{L'}{L} \rceil$. Then there exists an algorithm A' that achieves regret $\alpha^d \cdot r\left(\frac{T}{\alpha^d}\right)$ for learning an L'-Lipschitz function with symmetric loss in the d-dimensional case.*

*Proof.* Partition $[0,1]^d$ into $\alpha^d$ small cubes of edge length $1/\alpha$. For each such cube, if we expand the edge length by a multiple of $\alpha$, then any function that is $L'$-Lipschitz in the cube will become $L$-Lipschitz. So our new algorithm will just expand the edge length of all cubes by a multiple of $\alpha$ and run algorithm $A$ on each small cube separately.

Label the $\alpha^d$ small cubes $1$ through $\alpha^d$, and assume cube $i$ receives $T_i$ queries. We have $T = T_1 + \cdots + T_{\alpha^d}$. By Jensen's inequality, the new algorithm will have regret

$$\sum_{i=1}^{\alpha^d} r(T_i) \leq \alpha^d \cdot r\left(\frac{T}{\alpha^d}\right).$$

$\square$

**Lemma 15.** *Suppose an algorithm A achieves regret $r(T)$ for learning an 1-Lipschitz function with symmetric loss in the d-dimensional case. For any $L < 1$, let $\alpha = \lfloor \frac{1}{L'} \rfloor$, we have an algorithm that achieves regret at most $\frac{4}{\alpha} \cdot r(T) + O(1)$ for learning an L-Lipschitz function with symmetric loss in the d-dimensional case.*

*Proof.* When $\alpha \leq 4$, the bound follows trivially from just applying $A$. Now we assume $\alpha \geq 4$ and therefore $L \leq 1/4$. When the function is $\frac{1}{4}$-Lipschitz, we know the gap between the maximum value and the minimum value of the function in $[0,1]^d$ is at most $\frac{1}{4}$.

Now, similarly to the process in Algorithm 1, we do a binary search to find a small interval $Y$ such that $f([0,1]^d) \subset Y$, stopping when $\text{length}(Y)$ is no more than $\frac{4}{\alpha}$. This step results in at most $O(1)$ regret (since the length of $Y$ decreases by a constant factor every query). We can then multiply the function range by a factor $\frac{\alpha}{4}$ and run algorithm $A$ for the remainder of the rounds. This part has regret at most $\frac{4}{\alpha} \cdot r(T)$. This new algorithm has regret at most $\frac{4}{\alpha} \cdot r(T) + O(1)$ for learning an $L$-Lipschitz function with symmetric loss.

$\square$

# B  Omitted algorithms and proofs

### Proof of Theorem 2

*Proof.* We begin by proving some preliminary lemmas. We begin by showing that it is always the case that $f(X_j) \subseteq Y_j$.

**Lemma 16.** *Let $X_j$ be an interval in the partition maintained by Algorithm 1 (at some time t). Then for any $x \in X_j$, $f(x) \in Y_j$.*

*Proof.* Note that in the initial partition, all the $Y_j$'s start off equal to $[0,1]$, where this holds trivially. In addition, whenever we create new intervals (via bisection of old intervals), we initially set the range for the each of the new intervals equal to the range for the old itnervals. It therefore suffices to show that this property continues to hold whenever we update $Y_j$ in response to learning whether $y_t > f(x_t)$.

We update $Y_j$ in response to learning that $y > f(x)$ (or $\leq f(x)$) for some $x \in X_j$ and some $y \in Y_j$. Note that if $f(x) < y$, then by the Lipschitz property, for any $x' \in X_j$, $f(x') < f(x) + L|x' - x| <$

$y + L \cdot \text{length}(X_j)$. Likewise, if $f(x) > y$, then by the Lipschitz property, for any $x' \in X_j$, $f(x') > f(x) - L|x' - x| > y - L \cdot \text{length}(X_j)$. Letting $\ell_j = \text{length}(X_j)$, it follows that in the first case $f(X_j) \subseteq [0, y + L\ell_j]$ and in the second case, $f(X_j) \subseteq [y - L\ell_j, 1]$, and therefore the update rules preserve this property. □

We now show that if $h_j \geq 4L\ell_j$, then performing the update in lines 8-12 of the algorithm decreases $h_j$ by a factor of at least $3/4$ and at most $1/2$.

**Lemma 17.** *Let $h_j$ equal the length of $Y_j$ before the update in lines 8-12, and let $h'_j$ equal the length of $Y_j$ after the update in lines 8-12. Then if $h_j \geq 4L\ell_j$, $h'_j/h_j \in [1/2, 3/4]$.*

*Proof.* Recall that the guess $y_t$ is the midpoint of the interval $Y_j$. Regardless of whether $y_t > f(x_t)$ or $y_t \leq f(x_t)$, $h'_j = h_j/2 + L\ell_j$. It immediately follows that $h'_j/h_j \geq 1/2$. On the other hand, since $h_j \geq 4L\ell_j$, $h'_j/h_j = 1/2 + L\ell_j/h_j \leq \frac{3}{4}$. □

Finally, we show that it is always the case that (at the beginning of a turn) $h_j \geq 4L\ell_j$.

**Lemma 18.** *Let $X_j$ be an interval in the partition (at the beginning of some turn t). Then $\text{length}(Y_j) \geq 4L \cdot \text{length}(X_j)$, and $\text{length}(Y_j) \leq 8L \cdot \text{length}(X_j)$.*

*Proof.* Let $h_j = \text{length}(Y_j)$ and $\ell_j = \text{length}(X_j)$.

By the construction of the initial partition, this is true at the beginning of turn 1. We must show that whenever $h_j$ drops below $4L\ell_j$, bisecting the interval $X_j$ (as in lines 14-16) restores this property. But this is true, since by Lemma 17, we must still have that $h_j > 2L\ell_j$ after the update to $Y_j$. When $X_j$ is bisected to form intervals $X_{j_1}$ and $X_{j_2}$, $\ell_{j_1} = \ell_{j_2} = \ell_j/2$, but $h_{j_1} = h_{j_2} = h_j$, so it is true that $h_{j_1} > 4L\ell_{j_1}$. Similarly, since $h_j < 4L\ell_j$ right before a bisection, $h_{j_1} = h_j < 4L\ell_j = 8L\ell_{j_1}$, as desired. □

We now analyze the regret incurred by this algorithm. We say an interval $X_j$ in our partition has depth $r$ if it is the result of the bisection of an interval at depth $r - 1$ (where the initial intervals all have depth 0). Note that if $X_j$ is at depth $r$, then $\text{length}(X_j) \leq 2^{-r}/8L$. By Lemma 18, it follows that if $X_j$ is at depth $r$, then $\text{length}(Y_j) \leq 2^{-r}$.

We claim that an adversary can choose a point $x_t$ belonging to an interval $X_j$ of depth $r$ at most $O(1)$ times before our algorithm splits $X_j$ into two smaller intervals. To see why this is true, note that by Lemma 18, $Y_j$ starts off satisfying $\text{length}(Y_j) \leq 8L \cdot \text{length}(X_j)$, and we must bisect $X_j$ as soon as $\text{length}(Y_j) \leq 4L \cdot \text{length}(X_j)$. But by Lemma 17, $\text{length}(Y_j)$ shrinks by a factor of at least $3/4$ each time $x_t$ belongs to $X_j$. Since $(3/4)^3 < 1/2$, after at most 3 iterations, we must divide $X_j$ into two smaller intervals. Note that the total amount of regret incurred in turns where $x_t$ belongs to an interval $X_j$ of depth $r$ is at therefore at most $\text{length}(Y_j) \cdot O(1) = O(2^{-r})$.

Now, there are at most $O(L2^r)$ intervals of depth $r$, so the total regret contributed from intervals of depth $r$ is $O(L)$. Over the course of $T$ rounds (where, by this analysis you can be charged regret $O(2^{-r})$ at most $O(L2^r)$ times), this leads to a total regret of at most $O(L \log T)$. □

**Proof of Theorem 3 (Algorithm 3)**

We define the diameter $\text{diam}(X)$ of a subset $X \subseteq \mathbb{R}^d$ to be the maximum of $||x - y||_\infty$ over all pairs $x, y \in X$. Note that $\text{diam}([0,1]^d) = 1$.

*Proof.* We proceed similarly to the proof of Theorem 2. Note that, without loss of generality we can assume that $L = 1/8$ (so the initial partition just contains the box $[0,1]^d$). If we show a regret bound of $O(T^{(d-1)/d})$ for the case where $L = 1/8$, then Lemma 14 implies a regret bound of $O(L^d(T/L^d)^{(d-1)/d}) = O(LT^{(d-1)/d})$ in general.

As before, the following analogues of Lemmas 16, 17, and 18 hold (with the proofs carrying over essentially verbatim).

**Lemma 19.** *Let $X_j$ be an interval in the partition maintained by Algorithm 1 (at some time t). Then for any $x \in X_j$, $f(x) \in Y_j$.*

**Algorithm 3** Algorithm for learning a $L$-Lipschitz function from $\mathbb{R}^d$ to $\mathbb{R}$ ($d > 1$) under symmetric loss with regret $O(LT^{(d-1)/d})$.

---

1: Learner maintains a partition of $[0,1]^d$ into boxes (cartesian products of intervals) $X_j$.
2: Along with each interval $X_j$, learner maintains an associated range $Y_j \subseteq [0,1]$ such that if $x \in X_j$, $f(x) \in Y_j$.
3: Initially, learner partitions $[0,1]^d$ into $\lceil (8L)^d \rceil$ boxes $X_j$ with side lengths $\leq 1/8L$ and sets all $Y_j = [0,1]$.
4: **for** $t = 1$ to $T$ **do**
5:     Learner receives an $x_t \in [0,1]$ from the adversary.
6:     Learner finds $j$ s.t. $x_t \in X_j$. Let $\ell_j = \mathsf{diam}(X_j)$.
7:     Learner guesses $y_t = (\max(Y_j) + \min(Y_j))/2$.
8:     **if** $y_t > f(x_t)$ **then**
9:         $Y_j \leftarrow Y_j \cap [0, y_t + L\ell_j]$
10:     **else**
11:         $Y_j \leftarrow Y_j \cap [y_t - L\ell_j, 1]$.
12:     **end if**
13:     Let $h_j = \mathsf{length}(Y_j)$.
14:     **if** $h_j < 4L\ell_j$ **then**
15:         Bisect each side of $X_j$ to form $2^d$ new boxes $X_{j_1}$ through $X_{j_{2^d}}$. Set $Y_{j_k} = Y_j$ (for $1 \leq k \leq 2^d$).
16:     **end if**
17: **end for**

---

**Lemma 20.** *Let $h_j$ equal the length of $Y_j$ before the update in lines 8-12, and let $h'_j$ equal the length of $Y_j$ after the update in lines 8-12. Then if $h_j \geq 4L\ell_j$, $h'_j/h_j \in [1/2, 3/4]$.*

**Lemma 21.** *Let $X_j$ be an interval in the partition (at the beginning of some turn t). Then* $\mathsf{length}(Y_j) \geq 4L \cdot \mathsf{diam}(X_j)$*, and* $\mathsf{length}(Y_j) \leq 8L \cdot \mathsf{diam}(X_j)$*.*

We again define the notion of *depth* by saying that a box $X_j$ in our partition has depth $r$ if it is the result of the bisection (into $2^d$ parts) of a box at depth $r-1$ (where the initial boxes all have depth 0). Since the diameter of each box in a bisection is exactly half of that of the original box, if $X_j$ is at depth $r$, then $\mathsf{diam}(X_j) \leq 2^{-r}/8L$. By Lemma 18, it follows that if $X_j$ is at depth $r$, then $\mathsf{diam}(Y_j) \leq 2^{-r}$.

Similarly as to in the proof of Theorem 3, we can show that an adversary can choose a point $x_t$ belonging to a box $X_j$ of depth $r$ at most $O(1)$ times before our algorithm splits $X_j$ into $2^d$ smaller boxes. Note that the total amount of regret incurred in turns where $x_t$ belongs to a box $X_j$ of depth $r$ is at therefore at most $\mathsf{length}(Y_j) \cdot O(1) = O(2^{-r})$.

Now, there are at most $O(2^{rd})$ intervals of depth $r$ (since $L = 1/8$, so there is only 1 interval of depth 0), so the total regret contributed from boxes of depth $r$ is $O(2^{r(d-1)})$. Over the course of $T$ rounds (where, by this analysis you can be charged regret $O(2^{-r})$ at most $O(2^{dr})$ times), this leads to a total regret of at most $O(T^{(d-1)/d})$, as desired. $\qquad\square$

**Proof of Theorem 4 (Algorithm 4)**

*Proof.* We adapt the analysis from Theorem 3 (of Algorithm 3) to Algorithm 4. There are two primary differences between Algorithm 3 and Algorithm 4:

1. We stop following Algorithm 3 once the size of a range $Y_j$ corresponding to some box drops below $\tau = ((8L)^d/T)^{1/d+1}$. At this point, we always choose $\min(Y_j)$, so we are guaranteed not to overguess the value of $f(x_t)$, and the total loss from queries in such boxes is therefore at most $O(T\tau) = O((8LT)^{d/(d+1)}) = O((LT)^{d/d+1})$.

2. Unlike in Theorem 3, the total amount of regret incurred in turns where $x_t$ belongs to a box of depth $r$ is now $O(1)$ (instead of $O(\mathsf{length}(Y_j)) = O(2^{-r})$), since overguessing can

**Algorithm 4** Algorithm for learning a $L$-Lipschitz function from $\mathbb{R}^d$ to $\mathbb{R}$ ($d > 1$) under pricing loss with regret $O_d((LT)^{d/(d+1)})$.

---

1: Learner maintains a partition of $[0,1]^d$ into boxes (cartesian products of intervals) $X_j$.
2: Along with each interval $X_j$, learner maintains an associated range $Y_j \subseteq [0,1]$ such that if $x \in X_j$, $f(x) \in Y_j$.
3: Initially, learner partitions $[0,1]^d$ into $\lceil (8L)^d \rceil$ boxes $X_j$ with side lengths $\leq 1/(8L)$ and sets all $Y_j = [0,1]$.
4: **for** $t = 1$ to $T$ **do**
5:     Learner receives an $x_t \in [0,1]$ from the adversary.
6:     Learner finds $j$ s.t. $x_t \in X_j$. Let $\ell_j = \mathsf{diam}(X_j)$, and let $h_j = \mathsf{length}(Y_j)$.
7:     **if** $h_j \leq ((8L)^d/T)^{1/d+1}$ **then**
8:         Learner guesses $y_t = \min(Y_j)$.
9:     **else**
10:        Learner guesses $y_t = (\max(Y_j) + \min(Y_j))/2$.
11:        **if** $y_t > f(x_t)$ **then**
12:           $Y_j \leftarrow Y_j \cap [0, y_t + L\ell_j]$
13:        **else**
14:           $Y_j \leftarrow Y_j \cap [y_t - L\ell_j, 1]$.
15:        **end if**
16:        **if** $h_j < 4L\ell_j$ **then**
17:           Bisect each side of $X_j$ to form $2^d$ new boxes $X_{j_1}$ through $X_{j_{2^d}}$. Set $Y_{j_k} = Y_j$ (for $1 \leq k \leq 2^d$).
18:        **end if**
19:     **end if**
20: **end for**

---

lead to $\Theta(1)$ regret. This means that the total regret contributed from boxes of depth $r$ is $O((8L)^d 2^{rd})$, i.e. the number of intervals of depth $r$.

Now, since the length of $Y_j$ for a box $X_j$ of depth $r$ is at most $2^{-r}$, any box of depth at least $\log 1/\tau$ satisfies $\mathsf{length}(Y_j) \leq \tau$. The total number of boxes at depth at most $\log 1/\tau$ is $O((8L)^d 2^{d\log 1/\tau}) = O((8L/\tau)^d) = O((8LT)^{d/(d+1)}) = O((LT)^{d/d+1})$. Each of these boxes contributes $O(1)$ loss, so our total loss is $O((LT)^{d/d+1})$, as desired. $\qquad\square$

**Proof of Theorem 5**

*Proof.* We will mirror the analysis of Theorem 2.

Augment Algorithm 2 to keep track of a partition of $[0,1]$ into intervals $X_j$ along with associated ranges $Y_j$ for each interval $X_j$. Similarly as in Algorithm 1, we start with the partition into $\lceil 16L \rceil$ intervals $X_j$ of equal length $\leq 1/16L$. At any point in time, we define $Y_j$ via

$$ Y_j = \left[ \min_{x \in X_j} \max\{y \,|\, (x,y) \in S^-\}, \max_{x \in X_j} \min\{y \,|\, (x,y) \in S^+\} \right]. $$

Similarly as in Algorithm 1, once $\mathsf{length}(Y_j) < 6L\mathsf{length}(X_j)$, we will divide $X_j$ to form four new intervals of equal lengths.

By the definition of $Y_j$ (and $S^+$ and $S^-$), it is true that if $x \in X_j$, then $f(x) \in Y_j$. We will now prove the analogues of Lemmas 17 and 18.

**Lemma 22.** *Assume $x_t \in X_j$, and let $\ell_j = \mathsf{length}(X_j)$. Let $h_j$ equal the length of $Y_j$ before the update in lines 7-11, and let $h_j'$ equal the length of $Y_j$ after the update in lines 7-11. Then if $h_j \geq 6L\ell_j$, $h_j'/h_j \in [1/3, 5/6]$.*

*Proof.* Define $y_{mid} = (\min(Y_j) + \max(Y_j))/2$ (i.e., the guess Algorithm 1 would have made in this situation). We first claim that $y_t = (y^- + y^+)/2$ is close to $y_{mid}$, namely that $|y_{mid} - y_t| \leq L\ell_j$.

To do this, note that the boundary of $S^-$ and $S^+$ is composed of line segments of slope $L, -L$, and 0, so it is $L$-Lipschitz, and therefore $|y^- - \min(Y_j)| \leq \ell_j$, and $|y^+ - \max(Y_j)| \leq \ell_j$.

Now, if $y_t \leq f(x_t)$, then $\min(Y_j)$ increases (and $h_j$ decreases) by at least $|y_t - \min(Y_j)| - L\ell_j \geq |y_{mid} - \min(Y_j)| - |y_t - y_{mid}| - L\ell_j \geq h_j/2 - 2L\ell_j$. Since $h_j \geq 6L\ell_j$, this is at least $L\ell_j$, so $h'_j/h_j \leq \frac{5}{6}$. Similarly, $\min(Y_j)$ increases by at most $|y_t - \min(Y_j)| \leq |y_{mid} - \min(Y_j)| + |y_t - y_{mid}| \leq h_j/2 + L\ell_j$, so $h'_j/h_j \geq \frac{1}{3}$.

By symmetry, when $y_t > f(x_t)$, it is also true that $h'_j/h_j \in [1/3, 5/6]$. $\qquad\square$

**Lemma 23.** *Let $X_j$ be an interval in the partition (at the beginning of some turn $t$). Then* $\mathsf{length}(Y_j) \geq 6L \cdot \mathsf{length}(X_j)$, *and* $\mathsf{length}(Y_j) \leq 16L \cdot \mathsf{length}(X_j)$.

*Proof.* Let $h_j = \mathsf{length}(Y_j)$ and $\ell_j = \mathsf{length}(X_j)$.

By the construction of the initial partition, this is true at the beginning of turn 1. We must show that whenever $h_j$ drops below $6L\ell_j$, trisecting the interval $X_j$ (as in lines 14-16) restores this property. But this is true, since by Lemma 17, we must still have that $h_j > 2L\ell_j$ after the update to $Y_j$. When $X_j$ is divided into four equal intervals $X_{j_1}, X_{j_2}, X_{j_3}$, and $X_4$, $\mathsf{length}(X_{j_k}) = \ell_j/4$. On the other hand, for each $Y_{j_k}$, $h_j - 2L\ell_j \leq \mathsf{length}(Y_{j_k}) \leq h_j$ (since $\max(Y_j) - \max(Y_{j_k}) \in [0, L\ell_j]$ by the Lipschitz condition, and likewise $\min(Y_{j_k}) - \min(Y_j) \in [0, L\ell_j]$). It follows that $\mathsf{length}(Y_{j_k}) \geq 6L\mathsf{length}(X_{j_k})$. Similarly, since $h_j < 4L\ell_j$ right before a division, $\mathsf{length}(Y_{j_k}) \leq 16L\mathsf{length}(X_{j_k})$, as desired. $\qquad\square$

With these two lemmas, the proof of Theorem 2 applies to show that the midpoint algorithm achieves regret $O(L \log T)$.

$\qquad\square$

**Proof of Theorem 6**

*Proof.* Without loss of generality, we assume that $T = \alpha^d$ where $\alpha$ is an integer. Partition $[0, 1]^d$ into $\alpha^d$ small cubes of edge length $1/\alpha$. Pick $x_1, \ldots, x_T$ to be the center of all these cubes. Note that for any $i \neq j$, $\|x_i - x_j\|_\infty \geq 1/\alpha$. For each $i = 1, ..., T$, we pick $f(x_i)$ uniformly random to be $1/2 + L/(2\alpha)$ or $1/2 - L/(2\alpha)$ (independent of other $f(x_j)$'s). For any realized $f(x_1), ..., f(x_T)$, the function is $L$-Lipschitz as $\forall i \neq j$,

$$|f(x_i) - f(x_j)| \leq L/\alpha \leq L\|x_i - x_j\|_\infty.$$

The function values are also bounded in $[0, 1]$ since $L \leq T^{1/d} = \alpha$.

Since each $f(x_i)$ is independent from other $f(x_j)$'s, it's easy to see that any algorithm will have $L/(2\alpha)$ expected loss in each round. Therefore any algorithm will have expected regret at least

$$\frac{L}{2\alpha} \cdot T = \frac{LT}{2T^{1/d}} = \Omega(LT^{\frac{d-1}{d}}).$$

$\qquad\square$

**Proof of Theorem 7**

*Proof.* When $LT < 1$, the theorem statement becomes trivial. We now assume $LT \geq 1$.

Without loss of generality, we assume that $LT = \alpha^{d+1}$ where $\alpha$ is an integer. Partition $[0, 1]^d$ into $\alpha^d$ small cubes of edge length $1/\alpha$. Pick $x_1, \ldots, x_T$ to be the center of all these cubes such that each center is queried for $T^{1/(d+1)}/L^{d/(d+1)}$ rounds. We then have for any $i \neq j$ that $\|x_i - x_j\|_\infty \geq 1/\alpha$. For each center, we pick its function value uniformly random to be $3/4 + L/(4\alpha)$ or $3/4 - L/(4\alpha)$. For any realized $f(x_1), ..., f(x_T)$, the function is $L$-Lipschitz as $\forall x_i \neq x_j$,

$$|f(x_i) - f(x_j)| \leq L/(2\alpha) \leq L\|x_i - x_j\|_\infty.$$

The function values are also bounded in $[1/2, 1]$ since $L \leq \alpha$.

Consider the center of each cube. Each center is queried $T^{1/(d+1)}/L^{d/(d+1)}$ times and has value either $3/4 + L/(4\alpha)$ or $3/4 - L/(4\alpha)$. Since we are concerned with the expected regret of our algorithm over some distribution of functions, we can wlog assume the algorithm is deterministic. We divide the deterministic algorithms for this specific center into 2 cases:

- Case 1: the algorithm never query some value $> 3/4 + L/(4\alpha)$. Then with probability half, when the function value is actually $3/4 + L/(4\alpha)$, the algorithm has pricing loss $L/(2\alpha)$. In this case, the algorithm has expected regret at least $\frac{T^{1/(d+1)}}{L^{d/(d+1)}} \cdot \frac{L}{2\alpha} = \Omega(1)$ on one center.

- Case 2: the algorithm queries some value $> 3/4 - L/(4\alpha)$. Then with probability half, when the function value is actually $3/4 - L/(4\alpha)$, the algorithm has pricing loss $3/4 - L/(4\alpha)$ just in the round when it queries some value $> 3/4 - L/(4\alpha)$. In this case, the algorithm also has expected regret at least $\Omega(1)$ on one center.

To sum up, any algorithm will have expected regret at least $\Omega(1)$ on each center. Since there are $\alpha^d = (LT)^{\frac{d}{d+1}}$ centers, any algorithm will have expected regret at least $\Omega\left((LT)^{\frac{d}{d+1}}\right)$. $\qquad\square$

**Proof of Theorem 8**

*Proof.* We will sample a function randomly and we will show that any algorithm will have $\Omega\left(L\sqrt{\frac{\log T}{\log\log T}}\right)$ regret in expectation.

We first prove the case when $L \leq 1$. Let $m = \lfloor \log_2(T+1) \rfloor$. We will focus on the first $2^m - 1$ rounds. For the random function $f$, we fix $f(0) = f(1) = 1/2$. For function values on other values of $x_t$, we will sample $f(x_t)$ in round $t$ and make sure that there exists an $L$-lipschitz function that are consistent with the known $(x, f(x))$ pairs.

$$x_1 = 1/2$$

$$x_2 = 1/4 \qquad x_3 = 3/4$$

$$x_4 = 1/8 \qquad x_5 = 3/8 \qquad x_6 = 5/8 \qquad x_7 = 7/8$$

Figure 3: Construction of $x_t$ in the lower bound

Let's first fix the values of $x_1, ..., x_{2^m - 1}$ (as in Figure 3). We determine them in $m$ levels. In level $i$, we determine the value of $x_{2^{i-1}}, ..., x_{2^i - 1}$. For $j = 0, ..., 2^{i-1} - 1$, $x_{2^{i-1}+j}$ is set to $\frac{2j+1}{2^i}$.

Let's now define how to choose $f(x_t)$. We will proceed level by level. Set $\delta = 1/\sqrt{\log T \log\log T}$. For each $t$ in level $i$ (i.e. $t = 2^{i-1} + j$ for some $j \in \{0, ..., 2^{i-1} - 1\}$ and $x_t = \frac{2j+1}{2^i}$), we will define $f(x_t)$ based on its nearest defined locations: $f\left(\frac{j}{2^{i-1}}\right)$ and $f\left(\frac{j+1}{2^{i-1}}\right)$. Let the slope

$$d_t = \frac{f\left(\frac{j+1}{2^{i-1}}\right) - f\left(\frac{j}{2^{i-1}}\right)}{\frac{1}{2^{i-1}}}$$

and the average

$$m_t = \frac{1}{2} \cdot \left( f\left(\frac{j}{2^{i-1}}\right) + f\left(\frac{j+1}{2^{i-1}}\right) \right).$$

To choose $f(x_t)$, there are two cases:

- Case 1: If $|d_t| + L\delta > L$, $f(x_t) = m_t$.

- Case 2: Otherwise, $f(x_t) = m_t + \frac{L\delta}{2^i}$ with probability $1/2$, and $f(x_t) = m_t - \frac{L\delta}{2^i}$, with probability $1/2$.

It is not hard to check that after defining $f(x_t)$, there still exist a $L$-Lipschitz function that are consistent with the known $(x, f(x))$ pairs (in particular, this is true since every pair of neighboring $x_t$'s satisfies the Lipschitz constraint).

We will now show that in each level, a constant fraction of $x_t$ are defined via case 2. For each $x_t$ in level $i$, it has two children $x_{2t}$ and $x_{2t+1}$ in level $i+1$ (as in Figure 3). By the construction, we know that if $|d_t| + L\delta \leq L$, then one of $d_{2t}$ and $d_{2t+1}$ is $d_t + L\delta$ and the other one is $d_t - L\delta$. If we take a random path from root to leaves as in Figure 3, the sequence of $d_t$'s on the path behave like a random walk. Specifically this random walk has step length $L\delta$ and it stops when hits $L$ or $-L$. We know if the random walk does not hit $L$ or $-L$, it means that all the $f(x_t)$'s on the path is defined in case 2. Since the path has length $m = O(\log T)$ and we pick $\delta = 1/\sqrt{\log T \log \log T}$, we know that at least constant fraction of the paths never hit $L$ or $-L$. It follows that in each level, there are at least a constant fraction of $x_t$'s which are defined in case 2.

For any $x_t$'s in level $i$, if $f(x_t)$ is defined in case two, any algorithm will get expected loss at least $\Omega(L\delta/2^i)$ in round $t$. We know that in each level, at least constant fraction of $x_t$'s are defined in case 2. Therefore any algorithm will get expected regret

$$\Omega\left(\sum_{i=1}^{m} 2^{i-1} \cdot \frac{L\delta}{2^i}\right) = \Omega(L\delta m) = \Omega\left(L\sqrt{\frac{\log T}{\log \log T}}\right).$$

For the case when $L > 1$, if we directly use the above construction, the function value might exceed $[0, 1]$. Wlog we assume $L$ is an integer. Divide $x$-axis into $L$ regions: $[0, 1/L], [1/L, 2/L], ..., [(L-1)/L, 1]$. In $i$-th region, freshly sample a 1-Lipschitz function $f$ with $T/L$ locations according to the above procedure and construct function $f_i(x) = f\left(\left(x - \frac{i-1}{L}\right) \cdot L\right)$ for $x \in [\frac{i-1}{L}, \frac{i}{L}]$. Now consider function $f'$ which is defined as combination of $f_1, ..., f_L$: $f'(x) = f_i(x)$ for $x \in [\frac{i-1}{L}, \frac{i}{L}]$. It's not hard to see that $f'$ is $L$-Lipschitz. Also using the argument above, we get that any algorithm will have $\Omega\left(\sqrt{\frac{\log(T/L)}{\log \log(T/L)}}\right) = \Omega\left(\sqrt{\frac{\log T}{\log \log T}}\right)$ in each region $[(i-1)/L, i/L]$ for $i = 1, ..., L$. Since any algorithm does not learn anything about region $j$ from function values in region $i$ for $i \neq j$, any algorithm will have $\Omega\left(L\sqrt{\frac{\log T}{\log \log T}}\right)$ expected regret on function $f'$.

$\square$

**Proof of Lemma 10**

*Proof.* First, note that $\sin x$ and $\cos x$ are both 1-Lipschitz functions, and that the product of $d$ 1-Lipschitz functions bounded in $[-1, 1]$ is $d$-Lipschitz. From this, it follows that the $i$th component of $x(\theta)$ is $i$-Lipschitz.

Now, note that $F(\theta) = \langle x(\theta), v_\theta \rangle$ for some $v_\theta$. Now, for any $\theta'$,

$$
\begin{aligned}
F(\theta') &= \max_{v \in S} \langle x(\theta'), v \rangle \\
&\geq \langle x(\theta'), v_\theta \rangle \\
&= \langle x(\theta), v_\theta \rangle + \langle x(\theta') - x(\theta), v_\theta \rangle \\
&\geq F(\theta) - \|x(\theta') - x(\theta)\|_1 \\
&= F(\theta) - \sum_{i=1}^{d-1} |x(\theta')_i - x(\theta)_i| \\
&\geq F(\theta) - \sum_{i=1}^{d-1} i\|\theta' - \theta\|_\infty \\
&= F(\theta) - \frac{d(d-1)}{2}\|\theta' - \theta\|_\infty
\end{aligned}
$$

It follows that

$$F(\theta) - F(\theta') \le \frac{d(d-1)}{2}||\theta' - \theta||_\infty.$$

By symmetry, it is also the case that

$$F(\theta') - F(\theta) \le \frac{d(d-1)}{2}||\theta' - \theta||_\infty,$$

and therefore $F$ is $d(d-1)/2$-Lipschitz. □

**Proof of Lemma 13**

*Proof.* We show how to construct such a set of unit vectors in $\mathbb{R}^d$. By then taking the orthant with the largest number of elements of this set (and rotating this orthant appropriately), this then leads to a construction of such a set of unit vectors in $(\mathbb{R}^+)^d$ (at the cost of a factor of $2^d$).

Let $\mathbb{S}^{d-1}$ be the unit sphere in $\mathbb{R}^d$, and let $B(\alpha, u) \subset \mathbb{S}^{d-1}$ be the subset of the unit sphere of points that form an angle of at most $\alpha$ with $u$. Note that if $\mu$ is the boundary measure of $\mathbb{S}^{d-1}$, then simple calculus shows that $\mu(B(\alpha, u))/\mu(\mathbb{S}^{d-1}) = \Theta_d(\alpha^{(d-1)})$.

We will now construct our set of unit vectors iteratively in the following way. Assume our set already includes the unit vectors $u_1, u_2, \ldots, u_n$. Now choose $u_{n+1}$ to be any point in $S_{n+1} = \mathbb{S}^{d-1} \setminus \bigcup_{i=1}^n B(\alpha, u_i)$. Note that any point in $S_{n+1}$ forms an angle of at least $\alpha$ with all of the $u_i$. In addition, as long as $n\mu(B(\alpha, u_i)) \le \mu(\mathbb{S}^{d-1})$, $S_{n+1}$ is guaranteed to be nonempty. Therefore we can continue this procedure until $n \ge \mu(\mathbb{S}^{d-1})/\mu(B(\alpha, u_i)) = \Theta_d(\alpha^{-(d-1)})$. The result follows.

□