[Reviews · NeurIPS 2018]

Reviewer 1



This submission is pretty out of my expertise. I just carefully read the intro part and find it is well written and very clear.

Reviewer 2



Summary: This paper considers learning a Lipschitz function with binary feedback: the learner receives input x, outputs a prediction y and observes whether f(x) > y or f(x) < y. This is particularly relevant in the contextual pricing problem, where the valuation of a buyer is f(x), and the learner only observes whether the buyer buys or not. The work provides several algorithms to solve the cases of symmetric (= l1) loss, and "pricing loss", which corresponds to pricing problems. Performance guarantees are given for both 1-dimensional and multi-dimensional scenarios. Finally, authors also provide lower bounds that show (near) optimality. Specific comments: - What if there is additional noise in observations? For example, what if f(x) models a probability of sale, which is Lipschitz, but the outcome is in {0, 1}. It seems that it would be possible to implement this without changing the structures of the algorithms by much. - The main downside (somewhat hard to find) I found that the paper refers to the "full paper" a lot. I understand space is constraint and the work provides a lot of results crammed into 8 pages, but referring to the full paper multiple times annoyed me; maybe change full paper into appendix? I want to have the feeling I'm reading the actual paper. - On my first read I got a bit confused at sections 2.2 and 2.3 as the work switches back to the symmetric loss. Maybe the authors can consider switching the 2 (though I understand there are drawbacks to that as well). - Obviously, the paper is premised on the Lipschitzness of buyers; it would be great to hear some additional thoughts on what one can do to test this assumption. Relatedly, bounds become vacuous as the dimension d grows, that is inevitable. However, can more things be said if there is some sparsity structure imposed; say that the Lipschitz constant is small for but a few dimensions? - Great job with the illustrations, I found them very helpful. Overall: I think this is an excellent paper; the pricing problem is a particularly nice application for the results in this work. The paper is well-written and paints a complete picture of results. I rounded down to a score of an 8, only because I think the particular setting might not appeal to the widest audience, but what it does, it does very well: clear accept. -- Update after rebuttal: I think the author's addressed concerns adequately and remain of the opinion that this paper should be accepted.

Reviewer 3



This paper studies the problem of learning a Lipschitz function $f(x)$ from binary feedback. Two types of loss are considered, the symmetric ell-1 loss and the unsymmetric and discontinuous pricing loss. To minimize the losses, four algorithms are proposed. Alg. 1 maintains a set $S = { (X_j, Y_j) }$, where $X_j$ is a partition interval of feasible set $[0,1]$ and $Y_j$ contains the lower and upper bounds of $f(x)$. At each iteration, an adversary queries a point $x_t$. The learner response with estimate $y = mid(Y_j)$, where $X_j$ is the interval containing $x_t$. Base on the feedback, the learner shrinks $Y_j$ by intersecting itself with a set derived from Lipschitz property of $f(x)$. The iteration is finalized by splitting $X_j$ into two new intervals when the length of $Y_j$ can hardly be reduced. Alg. 2 encodes the upper bound and lower bound of $f(x)$ in sets $S+$ and $S-$ respectively. At each iteration, the learner receives a query $x_t$, composing an interval containing $f(x_t)$ by solving a linear optimization constrained on $S+$ and $S-$. And it uses the mid-point of the interval as the current guess. The $S+$ and $S-$ are updated based on the binary feedback. Alg. 3 and Alg. 4 are extensions of Alg. 1 to the case of multi-dimensional Lipschitz functions. The author proves that Alg. 3 and Alg. 4 can learn $f(x)$ with sublinear regret for the ell-1 loss and the pricing loss respectively. The two algorithm are optimal by giving the lower bound of Lipschitz learning problem. The problem is similar to learning a Lipchitz function from binary feedback with ell-1 loss and pricing loss by showing the optimality of their methods. The following questions need to be clarified. 1) All the proposed algorithms depend on the Lipschitz constant $L$. However, in practice, $L$ may not be satisfied. How to adapt the proposed methods when the condition is not meet ? 2) Alg. 2 only supports the one-dimensional learning problem. How to extend it to the multi-dimension learning problem ? 3) No empirical studies are performed. Though the author argues that the model can be used in dynamic pricing problem, no empirical evidence validates the performance.